biomedical engineering

web technologies, hardware/software interfaces, system architectures, integration and modelling

**Author for correspondence:**
Richard W. Bowman
e-mail: r.w.bowman@bath.ac.uk

# Simplifying the OpenFlexure microscope software with the web of things

Joel T. Collins[1], Joe Knapper[1], Samuel J. McDermott[2], Filip Ayazi[2], Kaspar E. Bumke[1], Julian Stirling[1] and Richard W. Bowman[1]

[1]Centre for Photonics and Photonic Materials, Department of Physics, University of Bath, Bath, UK
[2]Cavendish Laboratory, University of Cambridge, Cambridge, UK

JTC, 0000-0002-9382-7511; JK, 0000-0002-5519-1700;
SJM, 0000-0003-2736-5467; FA, 0000-0003-4521-9826;
KEB, 0000-0001-7603-0861; JS, 0000-0002-8270-9237;
RWB, 0000-0002-1531-8199

We present the OpenFlexure Microscope software stack which provides computer control of our open source motorised microscope. Our diverse community of users needs both graphical and script-based interfaces. We split the control code into client and server applications interfaced via a web API conforming to the W3C Web of Things standard. A graphical interface is viewed either in a web browser or in our cross-platform Electron application, and gives basic interactive control including common operations such as Z stack acquisition and tiled scanning. Automated control is possible from Python and MATLAB, or any language that supports HTTP requests. Network control makes the software stack more robust, allows multiple microscopes to be controlled by one computer, and facilitates sharing of equipment. Graphical and script-based clients can run simultaneously, making it easier to monitor ongoing experiments. We have included an extension mechanism to add functionality, for example controlling additional hardware components or adding automation routines. Using a Web of Things approach has resulted in a user-friendly and extremely versatile software control solution for the OpenFlexure Microscope, and we believe this approach could be generalized in the future to make automated experiments involving several instruments much easier to implement.

## 1. Introduction

We introduce the software stack developed for the OpenFlexure Microscope [1,2], an open-source, 3D-printed, and fully-automated

laboratory microscope. The microscope has been deployed around the world in a wide range of operating environments, posing unique challenges as it is used in almost equal measure by novice and expert microscopists. Some users need a simple graphical interface, while others require advanced scripting capabilities. While most microscopes are directly connected to a single computer through a variety of (often specialist) interfaces, we use an embedded Raspberry Pi and a web API (application programming interface) to enable local and remote control through internet protocol (IP) networks. Our software stack makes use of modern networking technologies and Web of Things standards to control the microscope and manage the data it generates. This architecture for network-connected microscopy has allowed the OpenFlexure Microscope to be used in a diverse range of settings without re-implementation or code duplication. Additionally, the extensibility and interoperability built in to our software architecture has allowed users to develop additional functionality and entirely new imaging modes without having to re-implement the more complex instrument control code. This article primarily describes the server application, the graphical web application client, and the Python scripting client. A full list of relevant software repositories is given at the end of the manuscript.

## 1.1. Web of things approach

In recent years, open web technologies have been widely adopted for controlling domestic hardware, in the 'Web of Things' (WoT) [3]. These network and web technologies have already addressed many of the problems faced by laboratories and have been proven robust, fast, and secure. Support staff deeply familiar with networking and web technologies are already in place at most research laboratories and teaching institutions. While prior work has introduced web technology into laboratories [4–8], these have lacked the comprehensive standardization required for true interoperability. Recently, however W3C, the primary international standards organization for the open web, have moved to standardise the Web of Things, with solid industry and community support [9].

Aligning our software stack with the W3C WoT Architecture [9] means that the only bespoke elements of our microscope control protocol are those specific to our application, i.e. the capabilities of our microscope. The way those capabilities are represented through the HTTP (hypertext transfer protocol) API is set by the WoT standard. In turn, our HTTP API can be described using the OpenAPI standard [10] which enables automatic generation of documentation. Tools exist that use OpenAPI descriptions to generate client (and server) code in a wide variety of programming languages. Crucially, these underpinning technologies are maintained by a large community of web developers, rather than a small number of microscopists. A major advantage of adopting the WoT standard is that code controlling multiple instruments for larger experiments can then be written in any modern language that supports web requests.

## 1.2. Existing microscope control systems

Splitting microscope control software into client and server applications is also done by Microscope Cockpit [11,12], though it relies on the `pyro` library [13] for communication between client and server. This Python-specific protocol makes it difficult to add clients for new languages. The most widely used microscope automation package is µManager [14–17], where 'device adapters' written in C++ provide code for specific devices, controlled by a common core. The MMCore API allows control from several languages, but the whole system runs as one process and device adapters are included as binary libraries. This requires all hardware to be connected to the same computer, and also means that only one top-level application can use it at a time (e.g. the whole system must be reinitialized to switch between the µ Manager interface and a Python script for a particular experiment). ImSwitch is a Python framework that can accommodate extremely complex microscopes, but must be used for everything from managing devices up to the user interface and so again is mutually exclusive with other control software [18].

The software stack as described in this manuscript is specific to the OpenFlexure Microscope, but has the potential to be generalized. Generalizing this WoT approach adds another competing solution to the previous paragraph, though we do not propose an all-in-one solution requiring the adoption of our server code and user interface. Instead, our hope is that the approach of splitting up instrument control with a documented, standards-compliant API can enable intercompatibility. For example, it would be possible to write a server to act as a bridge, allowing µ Manager device adapters to be controlled through an HTTP API. Similarly, a set of device adapters could enable WoT devices to be controlled by the existing MMCore system, and hence the wide variety of applications that use it. Crucially, the 'semantic typing' mechanism that allows WoT devices to declare their capabilities and

their compatibility with particular protocols enables this intercompatibility to be added gradually. Semantic types are not exclusive, so a particular device could be used from μ Manager or Cockpit or its own bespoke software, all through the same HTTP interface. We suggest that the taxonomy and object model provided by μ Manager could be an ideal basis for a set of semantic types for microscope components due to its wide adoption. In the long term, we hope that both open projects and proprietary hardware and software developers will adopt the WoT protocol as a common language, enabling more direct interoperability and eliminating the requirement to use any particular control system for a given experiment.

# 2. Architecture

## 2.1. Client–server model

A microscope's software stack generally includes low-level device control code, logic to integrate the hardware components into a useful instrument, automation for common tasks, a graphical interface for interactive control, and a way to script automated experiments. Treating the microscope as a WoT device naturally splits these functions into client applications (the graphical interface and scripting APIs) and a server handling the bulk of the logic and hardware control as depicted in figure 1.

This split between the server application and clients has several important advantages. First, it enables multiple client applications to connect to a microscope simultaneously. This allows a graphical interface to display a real-time camera feed, while a script controls sample manipulation, data acquisition, and analysis. The server also provides continuity, so devices need not be reinitialized every time an experiment script is started. Conversely, a single client can manage multiple microscopes simultaneously. This has allowed clinicians to image sample slides from several microscopes concurrently, dramatically increasing data acquisition throughput.

Separating the more complex server application from comparatively simple client applications makes it significantly easier to write client libraries in a broad set of languages. This means that microscope users can script experiments without having to re-implement the hardware control code in their language of choice, interface with a binary library, or learn a new language. It also ensures consistency between different languages and avoids duplicated effort, as most of the complexity is in the server application. Indeed, using the OpenAPI description, it is in principle possible to automatically generate clients in a wide range of languages, though these may not be as user-friendly as our current clients, which are written by hand.

## 2.2. Hardware architecture

Our server runs on a Raspberry Pi computer embedded in the microscope (figure 1). The server application handles communication with the sample translation stage, imaging camera, and any additional hardware, as well as logic for data management and additional functions such as tiled scans and autofocus. Running the server on an embedded computer ensures the hardware control code is running in a very well controlled environment. We automatically build and distribute an SD card image with a correctly configured operating system and our server and client applications pre-installed [19]. This eliminates the most troublesome aspect of distributing instrument control software, which is correctly installing and configuring the low-level drivers and libraries on an unknown computer, often clashing with system-level changes made to support other instruments connected to the same machine. The server application has been developed specifically for the Raspberry Pi hardware, and does not have built-in support for other camera types. The server has been adapted for use with other cameras and platforms, e.g. Jetson Nano [20], but is not yet intended as a multi-platform system. While the server currently runs on many platforms for development purposes, we only use it to control hardware when it runs on a Raspberry Pi.

Client applications can run on the embedded Raspberry Pi computer, making the microscope a stand-alone system to which keyboard, monitor, and mouse may be attached. More usually, client applications will run on other devices connected via a wired or wireless IP network, using the Raspberry Pi's ethernet and WiFi interfaces. By using IP networking, we enable control of multiple instruments with any external router or switch. Replacing different (often proprietary and expensive) connectors and adaptors [21] with commodity hardware makes experimental science more efficient and more accessible to resource-constrained scientists.

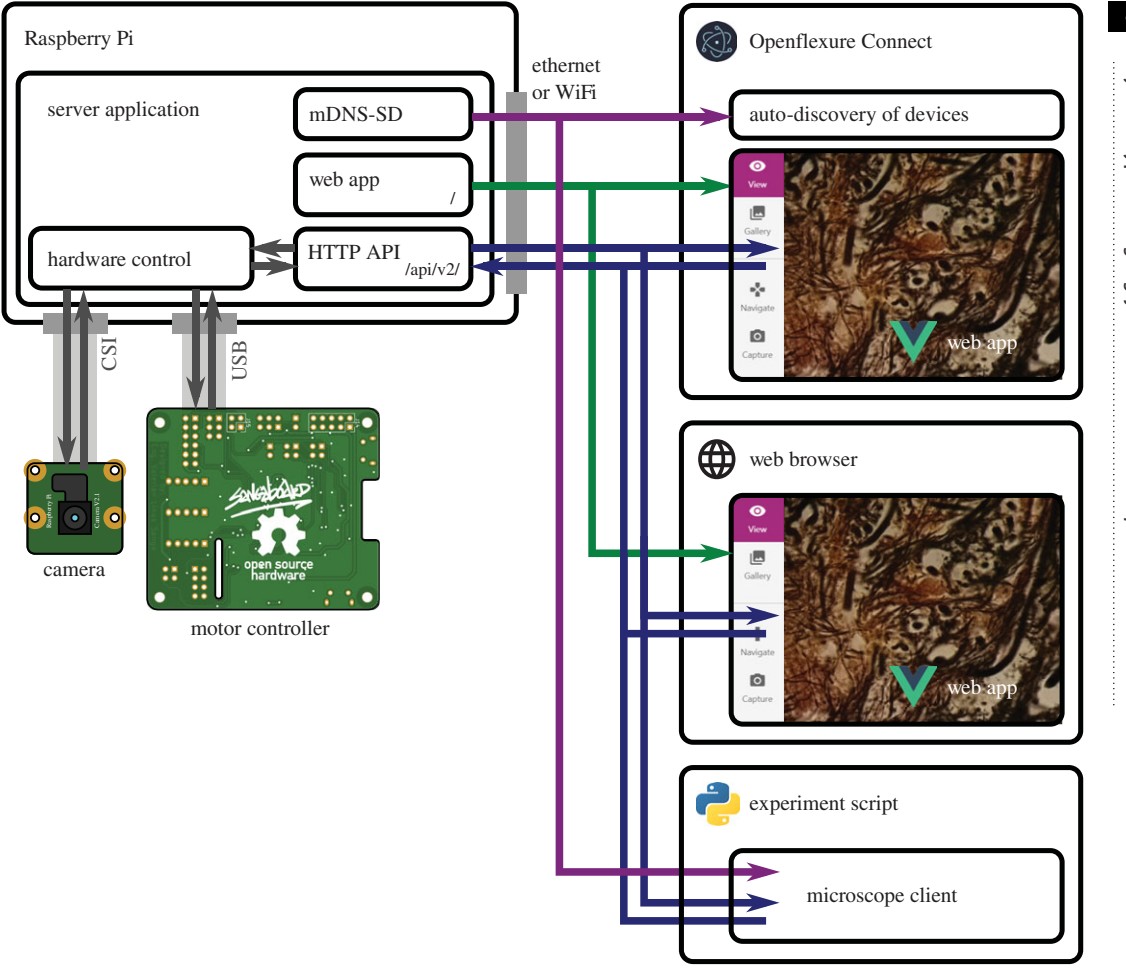

**Figure 1.** The structure of the OFM software and hardware. A server application (using Python and Flask) runs on the microscope's embedded Raspberry Pi. This includes code to control physical hardware, an HTTP API, and announcement of the microscope using mDNS to allow auto-discovery. The graphical interface is implemented as a web application using Vue.js, and is served as static files from the microscope server application. The OpenFlexure Connect application, built using the cross-platform Electron framework, automatically finds available microscopes using mDNS-SD and displays the web application in its own window. Alternatively, the web application can be displayed directly in a web browser, using the microscope's hostname or IP address. Experiment scripts can run at the same time, using a client module (at the time of writing, clients exist for Python and MATLAB). Clients may run anywhere on the network, including on the embedded computer if a stand-alone system is preferred.

The IP itself allows for high-speed, low-latency plug-and play communication. Fully remote control can be enabled using existing, well-established secure protocols such as SSH forwarding and VPN connections. Clients can automatically detect the server's IP address and capabilities via mDNS (multicast Domain Name System) [22], which is already used extensively by consumer WoT devices.

## 2.3. Server application and web API

The server application is written in Python, with much of the back-end code released as a separate library (Python-LabThings) [23]. This library uses the 'Flask' [24] web application framework, and includes various utilities to simplify thread-based concurrency, mDNS discovery, hardware synchronisation, and documentation generation. We also make use of standard scientific python libraries [25–28]. Our control code for the translation stage is also published separately, allowing its use outside of the microscope server [29], and we use a customized camera library to take manual control of the Raspberry Pi camera module [30].

Client–server interactions take place over HTTP. Our HTTP API follows the W3C Web of Things interaction model [9] that represents a device's capabilities through properties, actions, and events (collectively called 'interaction affordances'). Properties represent quantities that can be accessed

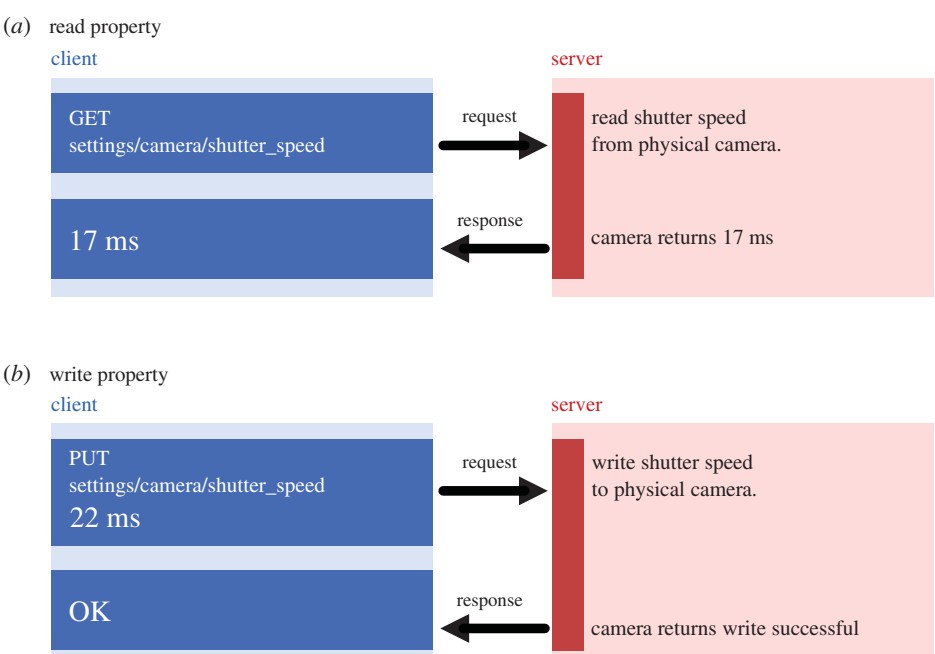

**Figure 2.** Simplified example of an HTTP request flow for reading and writing a 'property' resource. (*a*) Reading a property requires a GET request sent to the URL corresponding to the property to be read (camera shutter speed). The server then sends a response containing the properties value (17 milliseconds) in the response body. (*b*) Writing a property requires a PUT request sent to the same URL, containing the new value in the request body (22 milliseconds). The server then sends an 'OK' response confirming the operation's success.

immediately, such as the camera's exposure time or the current position of the stage. Actions cause the microscope to do something that may take some time, such as capturing an image or moving the stage. The WoT model defines the way interactions map onto HTTP requests and responses, following the principles of representational state transfer (REST) [31], a convention widely used across web service and WoT device APIs [32]. Most data are transferred using JavaScript Object Notation (JSON), a text based format that can serialize rich datatypes with informative, human-readable keys. Larger binary data such as images is usually not included in JSON objects, but is instead provided as a link to download the data in a suitable format, such as JPEG or PNG.

The widespread existing use of REST APIs means that users can interact with the OpenFlexure Microscope using existing standard libraries. We make use of the OpenAPI standard [10] to provide a machine-readable description of our API for automatic generation of interactive documentation [33]. This same mechanism can, in principle, be used to generate client code in a variety of languages. The key feature required to claim compatibility with the W3C WoT model is a description of the microscope's web API functionality in a standardised format [34]. This 'Thing Description' is a higher-level description of the API than OpenAPI, allowing more meaningful definitions relating to the capabilities of the device, rather than just a description of each exchange between client and server. Thing Descriptions may also be annotated with 'semantic types' that allow devices to fit into a taxonomy of device types and capabilities.

For example, the camera's exposure time is a property that can be read and written to as shown in figure 2. The stage's position is a read-only property; moving the stage is accomplished using an action. Actions do not have to complete instantaneously and thus require a mechanism to run in the background, after the initial web request has finished. Each interaction is represented by an HTTP URL or 'endpoint', and the HTTP method then describes the *type* of operation to be handled. For example, an HTTP GET request will read the value of a property, whereas an HTTP PUT request will write a new value to the property (figure 2).

### 2.3.1. Actions and concurrency

Many long-running tasks, such as acquiring large tile scans, must run in the background without blocking new API requests (e.g. monitoring the video feed or checking the progress of the task). Each

request and action is therefore handled by its own thread to allow concurrency. Access to physical hardware is carefully managed to avoid conflicting instructions by use of re-entrant locks. For example, if the translation stage is currently in the middle of a long movement, a request to move elsewhere will be denied until the initial move has completed, releasing the stage's lock. Clients can continue to interact with the microscope while an action is running, as long as they do not require a locked piece of hardware. This, for example, allows the live camera stream to be monitored during long-running experiments without users having to manually manage multiple threads.

Background tasks are handled automatically by the server. Whenever an 'action' is requested, the server will start the function in a new thread and immediately send a 'created' response back to the client, including the URL of an 'action resource' the client can poll to check its progress or final return value. In the future, we can improve efficiency by allowing the server to asynchronously *push* updates to clients without polling, for example using Server-Sent Events [35].

### 2.3.2. Extensions

Most automation and integration can be achieved using client-side code, which is simple to write and easy to debug. When the server code must be added to, for example to allow control of additional hardware components, Python code can be added to the server through our 'extension' mechanism. This also provides a way to make routines that are initially developed using client code to be made conveniently available to clients in any language, simplifying experiment scripts by providing higher-level actions. The main server application handles only the most basic microscope functionality: capturing images, moving the stage and managing device settings. A set of default extensions is included in the server package, providing higher-level capabilities including autofocus, scanning the stage to produce mosaics of images, and calibrating the camera and stage. Default extensions may be removed or replaced, allowing functionality to be customized on each microscope. Extensions are written as Python scripts that have direct access to Python objects that manage physical components comprising the microscope (e.g. camera and translation stage). Extensions can provide HTTP API endpoints and HTML interfaces that are displayed as part of the microscope's web app, and their capabilities are included in the microscope's built-in API documentation.

### 2.3.3. Security

The OpenFlexure Microscope software does not yet implement access control, and relies on network topology for security (e.g. the microscope should not be exposed to a large organizational network or to the internet). Secure remote access is already possible by using tools such as SSH to 'tunnel' connections, but in the future access control could be provided using standard security protocols that are provided for in the WoT standard, making the microscope more suitable for use on an insecure network.

## 2.4. Clients

Our primary client for the OpenFlexure Microscope is a web application included in the microscope's internal API server. This application provides a comprehensive graphical interface for the microscope, including a live stream of the camera, capture functionality, basic data management and full extension support. By developing the client as a browser-accessible web application, we are able to support many different operating systems without any additional code, while simultaneously drawing on the expertise brought by a large community of existing libraries.

The web application is accompanied by a desktop application (OpenFlexure Connect) handling device discovery and connection. The application finds and displays discovered microscopes using mDNS, as well as allowing manual connections and saving a list of commonly accessed microscopes. Upon connecting, the application loads the microscope's graphical user interface (figure 3). The interface is loaded from the microscope, so it can be customized to that microscope's capabilities and is guaranteed to be compatible with the API version in use.

Using a modular interface served by the microscope allows the client to only render user interface elements for enabled functionality. Server extensions are able to define new graphical interface components to be rendered within the main client application. For example, the interface to manage where images are stored is defined in the relevant extension. This mechanism is specific to our web application, rather than a general feature of the WoT standard. Interfaces are served through the

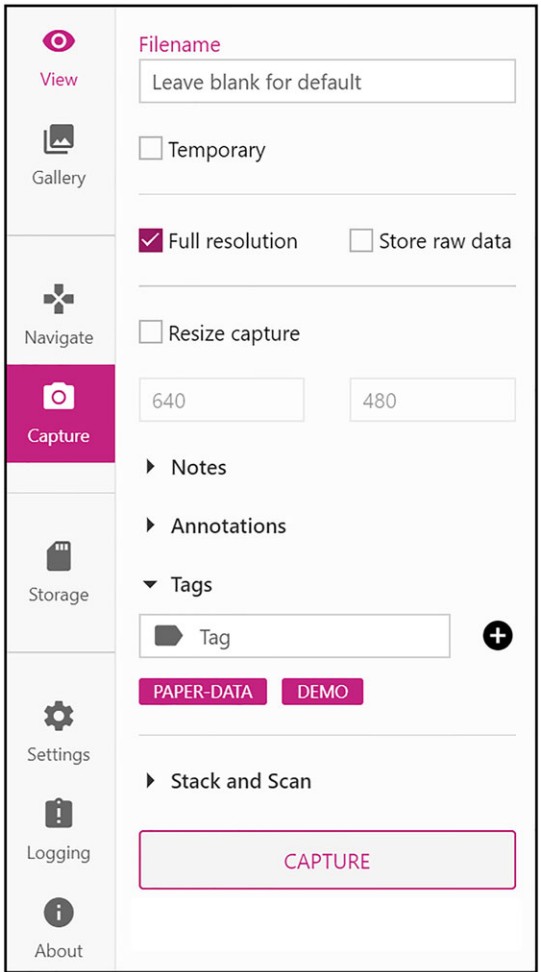

**Figure 3.** An example side pane from the OpenFlexure Connect microscope client. Using a web API for instrument communication allows the client to be written in any programming language. Using javascript we were able to create a cross-platform, responsive, and modern user interface using general purpose user interface frameworks.

HTTP API as HTML or JSON documents, linked to from the automatically generated description of the extensions.

For experiment scripting, we have created a Python client for the microscope [36,37] that converts the web API into native Python functions. Functionality provided by extensions is also mapped to Python functions automatically. This enables both scripted experiments and interactive sessions using, for example, Jupyter notebooks [38,39]. This lowers the barrier to entry for scripting microscopy experiments since many students and laboratory users are familiar with iPython notebooks, especially for data analysis. The ability to run the graphical client at the same time, e.g. to view the live video feed without any coding, further simplifies setting up and monitoring automated experiments. We have also created a client [40] with similar features. As the scripting clients are lightweight wrappers for the HTTP API, this involves a minimal amount of duplicated code.

The flexibility of the client–server architecture allows task or experiment specific interfaces to be created quickly. For example, a client for controlling the microscope with a USB game pad was developed [41]. This client is useful in remote field applications where a keyboard and mouse are not practical.

# 3. Implementation of microscope functionality

## 3.1. Live stream

Central to the microscope interface is the integrated live video display. The Raspberry Pi camera is supported by GPU firmware that provides accelerated JPEG compression of each video frame,

allowing us to serve a real-time Motion JPEG (MJPEG) live stream of the camera. The server starts a background thread on startup that records JPEG frames from the camera into a buffer. When clients connect to the stream, the server will begin sending a multi-part stream of frames taken from that buffer. A synchronization event is created for each client thread ensuring that clients never receive the same frame twice. As a new frame is read from the camera, the event for each client is set, at which point the response handler for each client will pass the frame onto the client, unset the event, and then wait for the event to be set again, dropping frames if necessary to ensure latency is minimized for clients not able to receive frames at the full rate. This system is based on the work of Miguel Grinberg [42], and is included in the Python-LabThings library. We use MJPEG in preference to more sophisticated video formats in order to minimize latency in the stream. The MJPEG format also makes it easy to extract or drop individual frames, and enables our fast autofocus algorithm.

## 3.2. Data collection

The Raspberry Pi camera can capture JPEG images and the raw 8 megapixel Bayer data from the camera sensor [30,43]. This raw Bayer data can be used for more advanced analysis, avoiding artefacts from gamma correction, demosaicing, and compression. The server records metadata about the state of the microscope at the time of capture (camera settings, stage position, calibration data and custom metadata added by the user), stored as a JavaScript Object Notation (JSON) formatted string in the 'UserComment' EXIF field. Captures are stored locally on the Raspberry Pi, either on the SD card or an available USB storage device, and can be listed (including metadata) and downloaded through the HTTP API. Separating the actions of capturing images and downloading them avoids the need to transfer large amounts of data over the network during time-critical phases of an experiment. The standard graphical client provides an gallery interface to view captured images and view their metadata.

As previously mentioned, multiple microscopes can be run in parallel to increase data throughput for a single operator. The greatest time saving can be achieved by setting microscopes to automatically scan over a large area, building a composite image of hundreds of overlapping fields of view (FOVs). The server has the option to perform such scans with movements, paths and capture types chosen by the user. Capture options and metadata are the same as individual captures, and individual images are saved, with metadata, as the scan runs [2]. Scans run as background tasks, so the microscope's video feed and position can be monitored as they run, and scans can be aborted without losing or corrupting images that are already acquired.

## 3.3. Autofocus

Due to the parallelogram-based mechanisms controlling the motion of the OFM translation stage, changes to the $x$-$y$ position move the sample over a sphere cap relative to the optics rather than a plane [1]. This necessitates an autofocus procedure which can be run reliably at each $x - y$ location in an automatic scan. As a typical diagnostic scan may require over 100 $x - y$ sites to be imaged, the software must focus rapidly while still being sufficiently reliable to not invalidate a large scan with any out-of-focus images.

A basic autofocus procedure captures a $z$-stack of images regularly spaced between points expected to be far above and below the focal point. At each height, an image is captured and converted to grey-scale. A Laplacian convolution is applied to the whole image, assigning higher values to areas of greater spatial brightness variance. These values are raised to the fourth power and summed over the image to provide a sharpness value. The translation stage is then returned to the $z$-position with the highest sharpness. This procedure is based on methods used previously to detect focused areas in an out-of-focus image [44], and while highly reliable, typically takes 10–20 s to complete, limited by capturing and image processing time.

A fast autofocus procedure uses the MJPEG preview stream as a metric of focus. By disabling bit rate control, the stream becomes a simple series of independent JPEG images each with identical compression settings. This JPEG compression uses the discrete cosine transform to describe blocks of the image [45], where each block is described using a superposition of the fewest discrete cosine functions possible, minimizing the storage space required. As focused images typically have sharper feature boundaries, the storage size of an MJPEG frame will peak when the sample is in focus. By tracking frame size and $z$-position as the objective moves through the focal point without stopping, the position of peak sharpness can be identified and returned to. On a sparse sample, blur in out-of-focus images can introduce information into blocks which would otherwise be empty. If this increase in the number of

blocks containing *some* non-zero information outweighs the reduction of blocks containing a large amount of information, the JPEG size may maximize away from focus. However, for feature-dense samples the size of a JPEG image can generally be used as a reliable measure of focus. This autofocus method has a greater positional resolution than the discrete steps of the simpler autofocus as MJPEG frame size can be tracked on-the-fly, reducing the time taken to less than 5 s. As autofocus is a rate-limiting step for scanning large samples, we have continued to refine the method in application-specific estensions, for example adding error correction to improve reliability when using oil immersion objectives [46].

## 3.4. Automatic calibration

### 3.4.1. Lens shading table

Due to chief ray angle compensation on the Raspberry Pi camera module's Sony IMX219 image sensor, the raw images captured by the microscope suffer from vignetting even when the sample is uniformly illuminated. However, flat-field correction allows us to recover uniform images in software [47]. We use a forked version [30] of the 'picamera' library [43] to access the lens shading table in the camera's GPU-based image processing pipeline, enabling us to correct for vignetting in both captured images and the real-time preview. A reduction in saturation at the edges of the image remains, but this can be corrected by post processing at the expense of higher noise in the image [47].

### 3.4.2. Camera-stage mapping

It is often convenient to move the microscope stage by a given displacement in pixels on the camera, but the axes and step size of the translation stage rarely align perfectly. We calibrate this relationship by moving back and forth along the stage's $x$- and $y$-axes in turn, analysing the resulting displacement in the image from the camera [48]. We combine the calibrations into a $2 \times 2$ affine transformation matrix that maps stage to camera coordinates. This is a similar approach to μManager's Pixel Calibrator plugin [17], but treating the camera's coordinate system as ground truth rather than the stage. We avoid hard-coded step sizes by moving the stage in gradually increasing steps, and measure mechanical backlash by comparing motion in opposite directions. This allows an intuitive click-to-move feature of the microscope's graphical interface, and will in the future be used when scanning and tiling images.

The same image analysis used for stage calibration is used as a two-dimensional displacement encoder, allowing for closed-loop sample translation and closed-loop scanning. This significantly reduces the error in each individual move, as well as ensuring that errors do not accumulate over the course of a large scan. Going forward, we will extend this functionality to include simultaneous location and mapping (SLAM) [49]. This will enable the creation of a map of the sample by comparing predictions based on commands sent to the motors to observations from the camera. This will enable accurate movements to features or areas of interest using the camera and estimated motor position.

## 4. Conclusion

The OpenFlexure Microscope software stack provides both graphical and scripted interfaces to the microscope's hardware capabilities, meeting the needs of users with a very wide range of technical abilities. Splitting the code into a server and multiple clients allows the microscope to be controlled as a stand-alone device or over a network, via an interactive application or through a script. Multiple clients can connect to the microscope at once, which reduces the need to duplicate the functions of the graphical interface in scripts that run particular experimental protocols. Using IP networks allows for multiple devices to access a microscope simultaneously, or for one computer to control multiple microscopes. Our standards-compliant HTTP API allows experiments to be scripted in almost any modern language on any operating system, using only standard libraries.

We have created a desktop application that makes it easy to find devices on a network, and display a graphical interface for the OpenFlexure microscope suitable for scientific, educational and clinical use. Extensions can add to or modify the graphical interface, to expose new capabilities or make it more suited to a particular purpose. We enable remote scripting of experiments with our Python and MATLAB clients, which can run alongside the graphical interface and integrate well with notebook-based programming.

The core architecture of our software is written as stand-alone libraries which are not specific to the microscope [23]. This makes the more generic aspects of our software, such as management of actions that run in threads, or the automatic generation of Thing Description and OpenAPI descriptions, available to other projects that wish to adopt a Web of Things approach. However, a key strength of our approach is that intercompatibility does not depend on the use of our library or framework, only on compliance to the W3C Web of Things standard, which is independent of any toolkit or framework.

Data accessibility. Pre-built software is available from our build server, linked from https://openflexure.org/projects/ microscope/. User and developer documentation for the server and the web application is available at https:// openflexure-microscope-software.readthedocs.io/, and developer documentation for LabThings is available at https://python-labthings.readthedocs.io/. All of our code described by this manuscript is available under open-source licenses, and repositories containing the development history of each project are available on GitLab and GitHub. The key repositories are: Server & web application: https://gitlab.com/openflexure/openflexure-microscope-server (includes Python server and Vue.js web application), Python client: https://gitlab.com/ openflexure/openflexure-microscope-pyclient, OpenFlexure Connect: https://gitlab.com/openflexure/openflexure-connect (Electron application to discover microscopes and display the web app), SD card image generator: https:// gitlab.com/openflexure/pi-gen, Python LabThings library: https://github.com/labthings/python-labthings. The current release of all of the repositories above is archived at https://dx.doi.org/10.5281/zenodo.5541935.

Authors' contributions. J.T.C. was primary software developer and drafted the manuscript. J.K. contributed to autofocus and testing code, and feedback on the software. J.S. contributed to design and review of the software and redrafted the manuscript. S.J.M., K.B. and F.A. contributed to the software, design and manuscript preparation. R.W.B. supervised the project and contributed to software design and development and manuscript preparation.

Competing interests. We declare we have no competing interests.
Funding. We acknowledge financial support from EPSRC (EP/R013969/1, EP/R011443/1) and the Royal Society (URF/ R1/180153, RGF/EA/181034).

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
