## [Peer Review File · Royal Society Open Science]

Review History

RSOS-211158.R0 (Original submission)

Review form: Reviewer 1

Is the manuscript scientifically sound in its present form?

Yes

Are the interpretations and conclusions justified by the results?

No

Is the language acceptable?

Yes

Do you have any ethical concerns with this paper?

No

Have you any concerns about statistical analyses in this paper?

No

Recommendation?

Major revision is needed (please make suggestions in comments)

Comments to the Author(s)

This manuscript proposes to make the “Web of Things” interface protocol central to microscope control in general, and in addition, describes the software developed to operate the OpenFlexure Microscope which is an “Open Source” microscope constructed with 3D-printed parts and freely available design documentation and support. There seems to be a general resistance to publishing manuscripts describing software architecture used in scientific experimentation, but I am highly supportive and see such publications not only as a “reward” for the work involved in architecting and writing the code, but also as an excellent way to communicate the design and reasons for the choices made so that others can benefit and expand on the work presented.

Nevertheless, I believe the manuscript can be strengthened by focusing either on the “Web of Things”, or on a description of the software created for the OpenFlexure microscope (which happens to use WoT protocols). In the current form, it is difficult or impossible to fully understand either. I am very intrigued by the proposed “Web of Things” interface protocol for microscope equipment, but I am unable to deduce from the manuscript how it works. Other than that it makes use of “modern networking technologies” (I presume it runs over TCP/IP?), and a reference to the W3C WoT recommendation (ref 7, which I did not read, and I do not think that a reader should first read this standard proposal to understand at least the basic protocols being used here for microscope control), I can find little about the actual protocol being used. Closest comes Fig. 2 that describes the methods to get and set properties in the hardware, but what is the “automatically generated description of the microscope’s web API functionality”? Is everything a property (including things like XY stage positions)? What are the mechanisms to get images over the client-server bridge (i.e. how are big binary blobs transferred)? Are the “Server extensions” that “define new graphical interface components” included in the WoT protocol, extensions of that protocol used by the OpenFlexure software, part of the WoT microscope interface protocol proposed by the authors, or something else entirely? Sections C and D leave me highly confused, whereas I had hoped to get away with a solid understanding of the protocol used between client and server. Expanding this section to methodically explain the protocol that client and server agree on would greatly strengthen the manuscript.

I am also unsure about the long term vision for WoT and microscope equipment. I guess that the authors propose that vendors of microscopes, stages, filter wheels and other microscope-related equipment would add an Ethernet port to their equipment that speaks the WoT protocol. What should a vendor like (for instance) Ludl, ASI, Prior, or Zaber do to make their XY stages compatible? How would their engineers find out what protocol to implement, and how would they know they did so correctly? How could a Zeiss AxioVert microscope with its industry standard CAN protocol be made compatible with the WoT protocol (seems a great proof of principle to me, a Raspberry Pi could very well function as the CAN-WoT bridge)? Or should each individual component in a microscope be addressable as an individual WoT device? What are the security implications of the WoT protocol? Are there safeguards build into the protocol or should the devices be secured at the network topology level? These, and many more, questions arise after giving the matter a small amount of thought, and if the authors really would like to posit the WoT interface as the desired interface for the industry to coalesce onto, there should be a decently coherent description and vision presented in the manuscript.

Alternatively, rather than focusing the manuscript on the “Web of Things” as the way forward for the microscopy industry with its “proprietary and legacy connector and protocols”, the authors could present their control software for the OpenFlexure Microscope (including their use of WoT protocols) and only in the Conclusion section discuss the possibilities to extend the WoT architecture to other microscope devices. I think that I would enjoy such an approach much more,

and believe it could actually make readers more enthusiastic about the idea. Reorganization of the manuscript (mainly rewriting abstract, introduction and conclusion) towards this end, in addition to better description of the protocols used, would in my opinion greatly improve the impact of this manuscript.

Review form: Reviewer 2

Is the manuscript scientifically sound in its present form?

Yes

Are the interpretations and conclusions justified by the results?

Yes

Is the language acceptable?

Yes

Do you have any ethical concerns with this paper?

No

Have you any concerns about statistical analyses in this paper?

No

Recommendation?

Accept with minor revision (please list in comments)

Comments to the Author(s)

In general, the tools here are impressive, though it is obviously not possible to test actual hardware control in this context. I think the work is deserving of publication, but I think it might be improved by clarification on a few points -

1) There are a lot of stack parts referenced here! The server, various clients (some specific to this microscope and some part of the labthings ecosystem), code for image capture, etc. Perhaps a figure here would clarify a) what code is "in-scope" (and will be eventually frozen/deposited), b) how everything interacts with everything else. This could possibly be built onto Figure 1, or at least structured similarly.

2) Can you address the degree to which it is NECESSARY to run the stack on a Raspberry Pi with a Raspberry Pi camera? Certainly parts of the stack can run elsewhere (I successfully built the server on my Mac per instructions), but it seems like other places in the stack like the camera capture, focus mechanism, etc may assume very particular hardware setups. This interrelates with my first point - it's not clear how specific certain aspects are intended to be for the OpenFlexure hardware vs general microscope control, which might be run on a Pi or something else entirely.

3) RE: integration with MicroManager and/or Cockpit- do either of those tools actually use similar REST APIs, and/or are there examples of tools integrating with them that way and/or is creation of such APIs on their official roadmaps? I think if so, some expansion here might be desired. Otherwise, at least for now, this IS "another competing device control system", though that certainly might not always be the case (I hope not!). Of course, desirability of this will a bit depend on points 1 and 2 - which parts of the stack are we discussing, and how generic are particular code pieces intended to be?

4) Minor point- is the server web app the same thing as Connect? I think it might be based on the figures, but it's not referred to as such in the server GitLab documentation - if these things are different, then how, and if not, should the server GitLab documentation refer to it as such?

4) Clearly quite a bit of effort has gone into the creation of the server Web app and in general it is very nice. It's certainly pretty user friendly, but I don't see documentation on any of the various repo(s) in terms of a "manual for end users" outside of the tour when the server is started - can you discuss if plans to create this (or promote it to higher prominence, if it does exist and I just missed it) exist? There is lots of developer documentation but not much that I can find for "if I set a microscopist in front of a scope running this software, what should they do?". It was hard for me to tell until I actually built the server that things like automated scanning, etc weren't just built into the various-language-clients but actually into the server webapp as well. Lack of something like this actually makes the tool seem LESS user friendly than it ultimately ends up being!

5) Minor point- is capturing non-JPEG data possible directly within the server webapp or only via client libraries (the Python library seems to have a distinction between `capture_image` and `grab_image`)? You state on page 7 that capturing the raw images is possible but it's not clear to me how a user would actually do it.

Decision letter (RSOS-211158.R0)

Dear Dr Bowman

The Editors assigned to your paper RSOS-211158 "Simplifying the OpenFlexure Microscope software with the Web of Things" have now received comments from reviewers and would like you to revise the paper in accordance with the reviewer comments and any comments from the Editors. Please note this decision does not guarantee eventual acceptance.

Please submit your revised manuscript and required files (see below) no later than 21 days from today's (ie 24-Aug-2021) date. Note: the ScholarOne system will 'lock' if submission of the revision is attempted 21 or more days after the deadline. If you do not think you will be able to meet this deadline please contact the editorial office immediately.

Please note article processing charges apply to papers accepted for publication in Royal Society Open Science (<https://royalsocietypublishing.org/rsos/charges>). Charges will also apply to

papers transferred to the journal from other Royal Society Publishing journals, as well as papers submitted as part of our collaboration with the Royal Society of Chemistry (<https://royalsocietypublishing.org/rsos/chemistry>). Fee waivers are available but must be requested when you submit your revision (<https://royalsocietypublishing.org/rsos/waivers>).

on behalf of Dr Michael Doube (Associate Editor) and R. Kerry Rowe (Subject Editor)
openscience@royalsociety.org

Associate Editor Comments to Author (Dr Michael Doube):

Associate Editor: 1

Comments to the Author:

Dear Dr Bowman,

We have received two excellent detailed reviews from experts in the field that are in principle supportive of publishing the manuscript (as am I), however, both reviewers complain of a lack of focus, theme and detail in the article and suggest some remedies. Please give the reviewers' comments your consideration and provide a revised manuscript for further assessment. Note that 'Major Revision' here means that changes to the text of the manuscript may significantly alter its meaning or its value to the community, but that changes to the engineering itself (apart from, potentially, user documentation) are not necessarily required.

Reviewer comments to Author:

Reviewer: 1

Comments to the Author(s)

This manuscript proposes to make the “Web of Things” interface protocol central to microscope control in general, and in addition, describes the software developed to operate the OpenFlexure Microscope which is an “Open Source” microscope constructed with 3D-printed parts and freely available design documentation and support. There seems to be a general resistance to publishing manuscripts describing software architecture used in scientific experimentation, but I am highly supportive and see such publications not only as a “reward” for the work involved in architecting and writing the code, but also as an excellent way to communicate the design and reasons for the choices made so that others can benefit and expand on the work presented.

Nevertheless, I believe the manuscript can be strengthened by focusing either on the “Web of Things”, or on a description of the software created for the OpenFlexure microscope (which happens to use WoT protocols). In the current form, it is difficult or impossible to fully understand either. I am very intrigued by the proposed “Web of Things” interface protocol for microscope equipment, but I am unable to deduce from the manuscript how it works. Other than that it makes use of “modern networking technologies” (I presume it runs over TCP/IP?), and a reference to the W3C WoT recommendation (ref 7, which I did not read, and I do not think that a reader should first read this standard proposal to understand at least the basic protocols being used here for microscope control), I can find little about the actual protocol being used. Closest comes Fig. 2 that describes the methods to get and set properties in the hardware, but what is the

“automatically generated description of the microscope’s web API functionality”? Is everything a property (including things like XY stage positions)? What are the mechanisms to get images over the client-server bridge (i.e. how are big binary blobs transferred)? Are the “Server extensions” that “define new graphical interface components” included in the WoT protocol, extensions of that protocol used by the OpenFlexure software, part of the WoT microscope interface protocol proposed by the authors, or something else entirely? Sections C and D leave me highly confused, whereas I had hoped to get away with a solid understanding of the protocol used between client and server. Expanding this section to methodically explain the protocol that client and server agree on would greatly strengthen the manuscript.

I am also unsure about the long term vision for WoT and microscope equipment. I guess that the authors propose that vendors of microscopes, stages, filter wheels and other microscope-related equipment would add an Ethernet port to their equipment that speaks the WoT protocol. What should a vendor like (for instance) Ludl, ASI, Prior, or Zaber do to make their XY stages compatible? How would their engineers find out what protocol to implement, and how would they know they did so correctly? How could a Zeiss AxioVert microscope with its industry standard CAN protocol be made compatible with the WoT protocol (seems a great proof of principle to me, a Raspberry Pi could very well function as the CAN-WoT bridge)? Or should each individual component in a microscope be addressable as an individual WoT device? What are the security implications of the WoT protocol? Are there safeguards build into the protocol or should the devices be secured at the network topology level? These, and many more, questions arise after giving the matter a small amount of thought, and if the authors really would like to posit the WoT interface as the desired interface for the industry to coalesce onto, there should be a decently coherent description and vision presented in the manuscript.

Alternatively, rather than focusing the manuscript on the “Web of Things” as the way forward for the microscopy industry with its “proprietary and legacy connector and protocols”, the authors could present their control software for the OpenFlexure Microscope (including their use of WoT protocols) and only in the Conclusion section discuss the possibilities to extent the WoT architecture to other microscope devices. I think that I would enjoy such an approach much more, and believe it could actually make readers more enthusiastic about the idea. Reorganization of the manuscript (mainly rewriting abstract, introduction and conclusion) towards this end, in addition to better description of the protocols used, would in my opinion greatly improve the impact of this manuscript.

Reviewer: 2

Comments to the Author(s)

In general, the tools here are impressive, though it is obviously not possible to test actual hardware control in this context. I think the work is deserving of publication, but I think it might be improved by clarification on a few points -

1) There are a lot of stack parts referenced here! The server, various clients (some specific to this microscope and some part of the labthings ecosystem), code for image capture, etc. Perhaps a figure here would clarify a) what code is "in-scope" (and will be eventually frozen/deposited), b) how everything interacts with everything else. This could possibly be built onto Figure 1, or at least structured similarly.

2) Can you address the degree to which it is NECESSARY to run the stack on a Raspberry Pi with a Raspberry Pi camera? Certainly parts of the stack can run elsewhere (I successfully built the server on my Mac per instructions), but it seems like other places in the stack like the camera capture, focus mechanism, etc may assume very particular hardware setups. This interrelates with my first point - it's not clear how specific certain aspects are intended to be for the

OpenFlexure hardware vs general microscope control, which might be run on a Pi or something else entirely.

3) RE: integration with MicroManager and/or Cockpit- do either of those tools actually use similar REST APIs, and/or are there examples of tools integrating with them that way and/or is creation of such APIs on their official roadmaps? I think if so, some expansion here might be desired. Otherwise, at least for now, this IS "another competing device control system", though that certainly might not always be the case (I hope not!). Of course, desirability of this will a bit depend on points 1 and 2 - which parts of the stack are we discussing, and how generic are particular code pieces intended to be?

4) Minor point- is the server web app the same thing as Connect? I think it might be based on the figures, but it's not referred to as such in the server GitLab documentation - if these things are different, then how, and if not, should the server GitLab documentation refer to it as such?

4) Clearly quite a bit of effort has gone into the creation of the server Web app and in general it is very nice. It's certainly pretty user friendly, but I don't see documentation on any of the various repo(s) in terms of a "manual for end users" outside of the tour when the server is started - can you discuss if plans to create this (or promote it to higher prominence, if it does exist and I just missed it) exist? There is lots of developer documentation but not much that I can find for "if I set a microscopist in front of a scope running this software, what should they do?". It was hard for me to tell until I actually built the server that things like automated scanning, etc weren't just built into the various-language-clients but actually into the server webapp as well. Lack of something like this actually makes the tool seem LESS user friendly than it ultimately ends up being!

5) Minor point- is capturing non-JPEG data possible directly within the server webapp or only via client libraries (the Python library seems to have a distinction between `capture_image` and `grab_image`)? You state on page 7 that capturing the raw images is possible but it's not clear to me how a user would actually do it.

===PREPARING YOUR MANUSCRIPT===

===PREPARING YOUR REVISION IN SCHOLARONE===

Author's Response to Decision Letter for (RSOS-211158.R0)

See Appendix A.

RSOS-211158.R0 (Revision)

Review form: Reviewer 1

Is the manuscript scientifically sound in its present form?

Yes

Are the interpretations and conclusions justified by the results?

Yes

Is the language acceptable?

Yes

Do you have any ethical concerns with this paper?

No

Have you any concerns about statistical analyses in this paper?

No

Recommendation?

Accept as is

Comments to the Author(s)

All my concerns have been addressed. It was a pleasure to read the manuscript.

I am a bit doubtful about the wisdom of the decision to run each Action in its own thread, rather than always running one thread per device, but that is an implementation detail, and apparently the current approach works.

Review form: Reviewer 2

Is the manuscript scientifically sound in its present form?

Yes

Are the interpretations and conclusions justified by the results?

Yes

Is the language acceptable?

Yes

Do you have any ethical concerns with this paper?

No

Have you any concerns about statistical analyses in this paper?

No

Recommendation?

Accept as is

Comments to the Author(s)

The manuscript is much improved- I thank the authors for their consideration of our comments and their work on addressing them. I am happy to recommend publication at this time.

A couple minor typos I did note -

P2 C1 L21 - "Adoptiong"

P3 C2 L48 - "OpanAPI"

Decision letter (RSOS-211158.R1)

Dear Dr Bowman,

It is a pleasure to accept your manuscript entitled "Simplifying the OpenFlexure Microscope software with the Web of Things" in its current form for publication in Royal Society Open Science. The comments of the reviewer(s) who reviewed your manuscript are included at the foot of this letter.

Please remember to make any data sets or code libraries 'live' prior to publication, and update any links as needed when you receive a proof to check - for instance, from a private 'for review'

URL to a publicly accessible 'for publication' URL. It is good practice to also add data sets, code and other digital materials to your reference list.

on behalf of Dr Michael Doube (Associate Editor) and R. Kerry Rowe (Subject Editor)
openscience@royalsociety.org

Associate Editor Comments to Author (Dr Michael Doube):

Comments to the Author:

I would like to thank the authors and reviewers for their constructive engagement over this manuscript, for which it has been a pleasure to act as Associate Editor.

Reviewer comments to Author:

Reviewer: 1

Comments to the Author(s)

All my concerns have been addressed. It was a pleasure to read the manuscript.

I am a bit doubtful about the wisdom of the decision to run each Action in its own thread, rather than always running one thread per device, but that is an implementation detail, and apparently the current approach works.

Reviewer: 2

Comments to the Author(s)

The manuscript is much improved- I thank the authors for their consideration of our comments and their work on addressing them. I am happy to recommend publication at this time.

A couple minor typos I did note -

P2 C1 L21 - "Adoptiong"
P3 C2 L48 - "OpanAPI"

Appendix A

Response to reviews

We are resubmitting our manuscript entitled “*Simplifying the OpenFlexure Microscope software with the Web of Things*” for your consideration. The manuscript has been improved following the suggestions of the referees to more clearly focus on the software as used by the OpenFlexure Microscope. We still believe the approach we have taken here has great potential to be generalised, but we agree with the referees that the generalisation is better left to a future publication, perhaps once there are more examples of instruments using the Web of Things standard. We have substantially modified the text and redrawn Figure 1, and believe we have addressed all of the issues raised by the referees.

We should also mention the addition of Kaspar Bumke to the author list; while the first version of this manuscript described the software before he had become strongly involved, he has made substantial contributions to software, maintenance, and manuscript preparation over the last several months and all authors agree that this contribution merits inclusion.

A full list of referee comments and our changes in response is included below. We hope that we have fully addressed the referees’ comments, and we are grateful for the constructive criticism that has led to the improved manuscript that is enclosed.

Reviewer 1

“

This manuscript proposes to make the “Web of Things” interface protocol central to microscope control in general, and in addition, describes the software developed to operate the OpenFlexure Microscope which is an “Open Source” microscope constructed with 3D-printed parts and freely available design documentation and support. There seems to be a general resistance to publishing manuscripts describing software architecture used in scientific experimentation, but I am highly supportive and see such publications not only as a “reward” for the work involved in architecting and writing the code, but also as an excellent way to communicate the design and reasons for the choices made so that others can benefit and expand on the work presented.

Nevertheless, I believe the manuscript can be strengthened by focusing either on the “Web of Things”, or on a description of the software created for the OpenFlexure microscope (which happens to use WoT protocols). In the current form, it is difficult or impossible to fully understand either.”

Both reviewers have raised this issue, and we have done our best to focus the paper on the microscope software stack. The introduction and conclusion have been substantially reworked, as has the abstract.

“

I am very intrigued by the proposed “Web of Things” interface protocol for microscope equipment, but I am unable to deduce from the manuscript how it works. Other than that it makes use of “modern networking technologies” (I

presume it runs over TCP/IP?), and a reference to the W3C WoT recommendation (ref 7, which I did not read, and I do not think that a reader should first read this standard proposal to understand at least the basic protocols being used here for microscope control), I can find little about the actual protocol being used. Closest comes Fig. 2 that describes the methods to get and set properties in the hardware, but what is the “automatically generated description of the microscope’s web API functionality”? Is everything a property (including things like XY stage positions)?”

We have improved our description of the protocol, clarifying how “properties” and “actions” are used in the software and specifying the two automatically generated documents (Thing Description and OpenAPI description) that describe the API.

“*What are the mechanisms to get images over the client-server bridge (i.e. how are big binary blobs transferred)?*”

We have added a sentence to specify that most data is exchanged as JSON, but that large binary blobs are generally transferred by first sending a download URL, then retrieving the binary data from that URL.

“*Are the “Server extensions” that “define new graphical interface components” included in the WoT protocol, extensions of that protocol used by the OpenFlexure software, part of the WoT microscope interface protocol proposed by the authors, or something else entirely?*”

We have clarified that this is specific to our web application, and not part of the WoT specification. While it would be a very useful extension to the standard, we don’t claim to have achieved that level of generality here.

“*Sections C and D leave me highly confused, whereas I had hoped to get away with a solid understanding of the protocol used between client and server. Expanding this section to methodically explain the protocol that client and server agree on would greatly strengthen the manuscript.*”

We have done our best to improve these sections, and our changes are highlighted in the enclosed “diff” document. In particular, we have done our best to make clear which aspects are governed by the WoT standard, and give a couple of illustrative examples that show what “properties” and “actions” look like.

We have also made the API documentation available without installing the microscope server¹. This always existed, but for technical reasons was only available from within the microscope software. The API documentation gives a very detailed description of all the actions and properties and should make it possible to understand the content of every transaction between client and server. While this is probably too detailed to be the right way to understand

¹<https://openflexure-microscope-software.readthedocs.io/en/master/api.html>

the protocol, we hope that, in tandem with the improved text, it leads to a more complete description of how everything works.

“
I am also unsure about the long term vision for WoT and microscope equipment. I guess that the authors propose that vendors of microscopes, stages, filter wheels and other microscope-related equipment would add an Ethernet port to their equipment that speaks the WoT protocol. What should a vendor like (for instance) Ludl, ASI, Prior, or Zaber do to make their XY stages compatible? How would their engineers find out what protocol to implement, and how would they know they did so correctly? How could a Zeiss AxioVert microscope with its industry standard CAN protocol be made compatible with the WoT protocol (seems a great proof of principle to me, a Raspberry Pi could very well function as the CAN-WoT bridge)? Or should each individual component in a microscope be addressable as an individual WoT device?
”

As we have now focused the paper on the OpenFlexure microscope, it’s probably a distraction to delve too deeply into this in the manuscript. However, I think the reviewer is definitely thinking along the same lines as us; instruments that use serial or CAN busses could be brought into the WoT ecosystem using a Raspberry Pi or other embedded computer as a ”bridge”, describing and exposing their capabilities to the network. Adding a network port to most modern instruments should be fairly simple; most of the microprocessors embedded in scientific devices are easily powerful enough to run an embedded web server. Demonstrating compatibility with the WoT standard can be done by validating the API description, and example projects are provided by the W3C and others on various platforms.

Achieving generic compatibility between, for example, different microscope stages requires them all to expose a compatible set of properties and actions. This is not standardised by the WoT protocol, and would require us to agree on a taxonomy of device types, with associated descriptions of the required capabilities. The ”semantic typing” mechanism defined by the WoT standard is very helpful here, as it allows multiple competing taxonomies to coexist, and devices can fit into several of them at the same time. We have explained this further in section I.B. There are already several competing taxonomies of microscope components embedded within the various microscope control systems we reference, and we do not propose to add another device hierarchy - our software is still specific to our microscope. We have suggested that in the first instance we might adopt micromanager’s taxonomy of devices and capabilities in our efforts to generalise, as this is the most widely-adopted standard in the community.

“
What are the security implications of the WoT protocol? Are there safeguards built into the protocol or should the devices be secured at the network topology level? These, and many more, questions arise after giving the matter a small amount of thought, and if the authors really would like to posit the WoT interface as the desired interface for the industry to coalesce onto, there should be a

decently coherent description and vision presented in the manuscript. ”

The WoT standard does provide for security including access control, but the OpenFlexure Microscope does not currently implement access control. Network topology is our main control at the moment, but as WoT technology becomes more widely used, we should adopt the standard methods of access control that are provided for in the standard. We have added a paragraph on security, as this is relevant to the microscope software as well as the more general WoT approach.

“ *Alternatively, rather than focusing the manuscript on the “Web of Things” as the way forward for the microscopy industry with its “proprietary and legacy connector and protocols”, the authors could present their control software for the OpenFlexure Microscope (including their use of WoT protocols) and only in the Conclusion section discuss the possibilities to extent the WoT architecture to other microscope devices. I think that I would enjoy such an approach much more, and believe it could actually make readers more enthusiastic about the idea. Reorganization of the manuscript (mainly rewriting abstract, introduction and conclusion) towards this end, in addition to better description of the protocols used, would in my opinion greatly improve the impact of this manuscript.* ”

We agree with the reviewer that this is a preferable approach for the current manuscript. We hope to posit our wider vision in a future work, accompanying a detailed implementation in the form of an improved LabThings library. However, we have endeavoured to focus the current manuscript on describing the software as it stands, specific to the microscope.

Reviewer 2

“ *In general, the tools here are impressive, though it is obviously not possible to test actual hardware control in this context. I think the work is deserving of publication, but I think it might be improved by clarification on a few points -*
1) *There are a lot of stack parts referenced here! The server, various clients (some specific to this microscope and some part of the labthings ecosystem), code for image capture, etc. Perhaps a figure here would clarify a) what code is “in-scope” (and will be eventually frozen/deposited), b) how everything interacts with everything else. This could possibly be built onto Figure 1, or at least structured similarly.* ”

We have redrawn Figure 1 with this in mind, and we believe it does as good a job as possible of showing how the many parts fit together.

“ 2) *Can you address the degree to which it is NECESSARY to run the stack on a Raspberry Pi with a Raspberry Pi camera? Certainly parts of the stack can run elsewhere (I successfully built the server on my Mac per instructions), but*

it seems like other places in the stack like the camera capture, focus mechanism, etc may assume very particular hardware setups. This interrelates with my first point - it's not clear how specific certain aspects are intended to be for the OpenFlexure hardware vs general microscope control, which might be run on a Pi or something else entirely. ”

We have clarified in the section on hardware architecture that the software presented here is specific to our microscope, and the Raspberry Pi architecture. It has been adapted for other platforms and we have mentioned and referenced this, but generalisation as a multi-platform control system is for the future. We have clarified that it does run on other platforms, but only for development purposes.

“ 3) RE: integration with MicroManager and/or Cockpit- do either of those tools actually use similar REST APIs, and/or are there examples of tools integrating with them that way and/or is creation of such APIs on their official roadmaps? I think if so, some expansion here might be desired. Otherwise, at least for now, this IS "another competing device control system", though that certainly might not always be the case (I hope not!). Of course, desirability of this will a bit depend on points 1 and 2 - which parts of the stack are we discussing, and how generic are particular code pieces intended to be? ”

We have focused the manuscript on the software stack for our particular microscope, as per both reviewers' suggestions. We have clarified that the software stack discussed in this paper is not (yet) generic, and have also added a list of relevant repositories within the data access statement at the end of the manuscript.

“ 4) Minor point- is the server web app the same thing as Connect? I think it might be based on the figures, but it's not referred to as such in the server GitLab documentation - if these things are different, then how, and if not, should the server GitLab documentation refer to it as such? ”

The two are separate - Connect is the electron application responsible for discovering microscopes on the network, and displaying the web application in a window. The web application refers to the contents of that window. From the user's perspective there is no distinction between the two when Connect is being used. As a historical note, the web application was at one point integrated with the predecessor to OpenFlexure Connect, though this has not been the case for some time and we have done our best to remove any references from the documentation.

We have clarified both the text and Figure 1, and think this is now much better described.

“ 4) Clearly quite a bit of effort has gone into the creation of the server Web app and in general it is very nice. It's certainly pretty user friendly, but I don't

see documentation on any of the various repo(s) in terms of a "manual for end users" outside of the tour when the server is started - can you discuss if plans to create this (or promote it to higher prominence, if it does exist and I just missed it) exist? There is lots of developer documentation but not much that I can find for "if I set a microscopist in front of a scope running this software, what should they do?". It was hard for me to tell until I actually built the server that things like automated scanning, etc weren't just built into the various-language-clients but actually into the server webapp as well. Lack of something like this actually makes the tool seem LESS user friendly than it ultimately ends up being! ”

This has been on our to-do list for some time, but has never reached the top! With thanks to the reviewer for the prompt to do this, our readthedocs site now includes a section on the web app aimed at users. This includes some screenshots and lists most of the built-in features - and we agree with the reviewer that it helps potential users evaluate the software before they have actually installed it. We have added a link to documentation in the data access statement.

“ 5) *Minor point- is capturing non-JPEG data possible directly within the server webapp or only via client libraries (the Python library seems to have a distinction between capture_image and grab_image)? You state on page 7 that capturing the raw images is possible but it's not clear to me how a user would actually do it.* ”

Data is only saved by the web app as JPEG images, however it is possible to embed raw Bayer data in those JPEG images, to allow later extraction and analysis of the raw data. This is done by checking the “Store raw data” box on the capture tab (and is now described in the web application documentation).